# Understanding the Patient Experience of Receiving Clinically Actionable Genetic Results from the MyCode Community Health Initiative, a Population-Based Genomic Screening Initiative

**DOI:** 10.3390/jpm12091511

**Published:** 2022-09-15

**Authors:** Anna Baker, Kasia Tolwinski, Jamie Atondo, F. Daniel Davis, Jessica Goehringer, Laney K. Jones, Cassandra J. Pisieczko, Amy C. Sturm, Janet L. Williams, Marc S. Williams, Alanna Kulchak Rahm, Adam H. Buchanan

**Affiliations:** 1Department of Psychology, Bucknell University, Lewisburg, PA 17837, USA or; 2Department of Psychology, Clemson University, Clemson, SC 29634, USA; 3Biomedical Ethics Unit, McGill University, Montreal, QC H3A 0G4, Canada; 4Department of Genomic Health, Geisinger, Danville, PA 17822, USA or; 5Department of Bioethics, Geisinger, Danville, PA 17822, USA; 6Heart and Vascular Institute, Geisinger, Danville, PA 17822, USA; 723andMe, Sunnyvale, CA 94086, USA

**Keywords:** genomic screening, biobank, genetic testing, psychological outcomes, patient experience

## Abstract

Understanding unselected individuals’ experiences receiving genetic results through population genomic screening is critical to advancing clinical utility and improving population health. We conducted qualitative interviews with individuals who received clinically actionable genetic results via the MyCode© Genomic Screening and Counseling program. We purposively sampled cohorts to seek diversity in result-related disease risk (e.g., cancer or cardiovascular) and in personal or family history of related diseases. Transcripts were analyzed using a two-step inductive coding process of broad thematic analysis followed by in-depth coding of each theme. Four thematic domains identified across all cohorts were examined: process assessment, psychosocial response, behavioral change due to the genetic result, and family communication. Coding of 63 interviews among 60 participants revealed that participants were satisfied with the results disclosure process, initially experienced a range of positive, neutral, and negative psychological reactions to results, adjusted positively to results over time, undertook clinically indicated actions in response to results, and communicated results with relatives to whom they felt emotionally close. Our findings of generally favorable responses to receiving clinically actionable genetic results via a genomic screening program may assuage fear of patient distress in such programs and guide additional biobanks, genomic screening programs, and research studies.

## 1. Introduction

Clinically driven genetic testing (i.e., testing based on a clinical indication, hereafter called clinical testing) occurs when a patient’s medical or family history raises concerns about inherited etiology for that history. However, genomic analysis of populations (i.e., results not from testing informed by a clinical indication, hereafter called population screening) is increasingly occurring in research and clinical settings [1,2], raising questions about the psychological impacts of disclosing clinically actionable genomic information discovered in these situations.

The psychological response to clinical testing has been largely positive, with no significant increase in anxiety and depression across multiple disease groups [3,4,5,6]. Further, there is a recognition that patients perceive genetic information as important for clinical decision-making [5]. However, increased anxiety after clinical testing has been reported in those with no personal history of the condition [7], fueling ongoing concern for population screening. Recent research on the psychological impact of secondary findings from clinical testing found that participants experienced no adverse psychological effects and that they shared results with relatives [3,8]. Thus, we sought to determine whether these psychosocial and family communication responses to clinical testing and secondary findings also surface among participants in population screening.

In 2019, the American College of Medical Genetics and Genomics (ACMG) updated guidance regarding secondary findings from clinical exome or genome sequencing [9,10], noting that the secondary finding gene list was not validated for general population screening and recommending further study on penetrance and utility in the research setting [11]. Understanding the impact of receiving results on psychological outcomes in the context of population screening is a critical component of the assessment of clinical utility. Importantly, if negative responses such as increased anxiety are found, the behavioral implications could adversely impact the personal utility of population screening. Research has shown that individuals with high levels of health-related anxiety, despite actual risk, are more likely to negatively interpret information, experience negative behavioral responses, regard themselves to be at greater risk, and fail to engage in protective strategies such as monitoring of symptoms [12]. Uncertainty about the genetic results’ clinical and familial implications can lead to a variety of emotional responses, which may impact individuals’ capacity and interest in seeking information that could resolve the uncertainty [13]. Further, uncertainty about the results’ implications can be associated with individuals perceiving the results to have lower value, fewer health benefits, and higher potential for harm [6,14]. In addition, if individuals perceive genetic findings as fixed attributes, this can lead to a reluctance to engage in risk-mitigating behavior [15].

Thus, to understand reactions to genomic information generated from population screening and the potential downstream utility for health, it is important to examine both the reported psychological reactions and the behaviors taken by individuals who receive genomic information in this context. Here, we report the lived experiences of individuals after receiving genomic information from a research project returning results in an unselected health system cohort from 2016–2018. Data on these post-disclosure experiences were collected from semi-structured interviews across situations that are likely to occur as population screening continues to be studied and implemented.

## 2. Materials and Methods

### 2.1. Setting

The Geisinger MyCode^®^ Community Health Initiative (MyCode) is a precision health research project enrolling participants from a rural, integrated health system serving central and northeastern Pennsylvania [16]. MyCode collects participant biospecimens and clinical data to investigate the genomic underpinnings of health and disease and thereby make advances in disease prevention and treatment [16]. In 2013, after ethical review and stakeholder input, project leadership made the decision to return clinically actionable results—due, in part, to the fact that participants “overwhelmingly favored the return of results” [17,18]. Research exome sequence data are reviewed for expected pathogenic and likely pathogenic variants [19] in a list of genes similar to the ACMG Secondary Findings v2.0 list [10]. These variants are clinically confirmed and reported to participants and their primary care physicians [18,20]. The disclosure process includes uploading the genetic result into the electronic health record (EHR), initial contact disclosing that clinically actionable information has been found, recommending genetic counseling for more detailed discussion of results’ clinical and familial implications, and offering cascade testing to MyCode participants’ at-risk relatives [20].

### 2.2. Interview Process

We developed an interview process to elicit the lived experiences of individuals who have received genetic results via MyCode. Semi-structured interview guides were developed using an experiential phenomenological approach [21]. Interview questions were informed by literature on reactions to genetic information after clinical testing, by expert opinion on individuals’ possible reactions to population genomic screening, and through consultation with local MyCode community and clinician stakeholder groups.

All interview guides included core questions designed to probe interviewees’ experience of receiving results, psychological response to that result, understanding and sharing of information with family, and healthcare decision-making related to recommendations specific to the result (Table 1). Interview guides were adapted to include condition- or topic-specific questions for each of the study cohorts described below and supplemented with additional questions specific to the cohort (e.g., medication use for the FH cohort, additional probes on experience for those whose family history was negative for disease associated with the genetic result, and questions about the learning of the familial risk in the cascade cohort).

### 2.3. Sampling Strategy

We used a purposive sampling strategy to ensure capture of diverse experiences and reactions to results in different MyCode participants. This study analyzed the semi-structured interviews with MyCode participants drawn from four specific groups: (1) individuals who were two months post-disclosure of results (PEX cohort); (2) participants with no family history of the condition associated with the returned result (hereditary breast and ovarian cancer syndrome (HBOC) or hypertrophic cardiomyopathy (HCM)) (FHxNeg cohort); (3) individuals who received a result for familial hypercholesterolemia (FH) (FH cohort) [22]; and (4) individuals who learned about their risk from a family member who received a MyCode result then pursued cascade testing (Cascade cohort). We purposively sampled groups with different characteristics (e.g., with and without relevant family history) to allow for the possibility that their experience could differ from that of individuals we had already interviewed.

### 2.4. Data Analysis

Study personnel compiled episodic summaries for each interview [23,24]. These summaries, along with the interview guides and existing literature, facilitated initial codebook construction. Interview transcripts were coded and analyzed using an abductive approach [25] which combines an inductive or grounded method to find emergent themes with a deductive method that compares those findings to existing literature and frameworks. Coding was conducted in two rounds. The first round identified general themes seen across all sample groups and was conducted by two study personnel who coded by consensus [26]. The second round included in-depth coding and analysis of each thematic domain; only codes with an inter-rater reliability of 80–100% [27] were included in the final codebooks. Themes from the final round of coding were summarized and illustrated using exemplar quotes, according to analysis and reporting standards of qualitative research [21,28,29].

## 3. Results

Sixty MyCode participants with a genetic result completed sixty-three interviews (three individuals completed two interviews each through participation in two study cohorts). Table 2 summarizes participants’ characteristics by cohort, including the numbers of participants with germline cancer or cardiovascular disease risk. Overall response rate was 38%; participation rate among potential participants reached was 59%. Median age across all cohorts was 58 (range 27–86); 60% of participants (n = 36) across cohorts were female.

Four thematic domains were found to be most relevant to patient experience in every sampling group: (1) disclosure process assessment; (2) psychological response; (3) behavior change due to the result; and (4) family communication about the result. All four thematic domains were found in all interviews.

### 3.1. Disclosure Process Assessment Domain

This domain captured interviewees’ experience with and feelings about participating in MyCode as well as feedback about the MyCode results disclosure process.

Most interviewees expressed favorable views of the MyCode project overall and had “no regrets” and only positive feelings about their involvement and the process used to return genetic results. Some interviewees found the summary letter (provided to all participants receiving a result) explaining their genetic condition helpful and stated that it led to desired health-related behaviors, while others reported the genetic information in the letter was difficult to understand:


*“I used the letter. I thought [it was] very useful, and then other reference information I thought was good because you don’t always remember… everything from the appointment” (PEX)*



*“Well, like I said, I am not a very medical person, and you know, most of it was like reading Greek to me.” (FHxNeg)*


### 3.2. Psychological Response Domain

This domain captured how interviewees described their initial response to their MyCode result and how they felt at the time of the interview. Interviewees reported a range of responses—from positive to neutral to negative—regarding receipt of the genetic result, their emotional “management” of the initial result, and the subsequent evolution of their feelings.

Initial negative responses reported by interviewees included feelings of alarm or fear, guilt, and concern about what the genetic information meant for their future health and the health of their families. Some used descriptors such as “devastating”, “awful”, and “shocking”, and that they “felt bitter.” Others indicated a sense of resignation. Many reported that due to their family history (whether objectively specific to the condition or perceived as relevant by the interviewee), they felt a higher level of concern or unease about the result’s implications. These included concern about increased risk for specific gene-related issues and about the procedures they might need in the future, anxiety about how to communicate the result to family members, and worry about insurance and other costs associated with their own or family members’ medical care:


*“… but we were wondering with my older [child] who works for a company, if he found out he had this kind of result, would that affect his healthcare coverage.” (FHxNeg)*


Interviewees without a known family history relevant to the condition—a history that might have prepared them for the result—reported feeling “shocked,” “surprised,” or, as in one case, like they were in a “twilight zone.” Some interviewees also had existing health problems and felt that the results added to the uncertainty of their health status or reported a feeling that their body was “wearing down” or failing them:


*“I was kind of overwhelmed because I was having other issues, and now it’s like, is that what was causing my other issues or…Yeah, I don’t know which came first. The chicken or the egg, you know.” (PEX)*


Regardless of cohort, some interviewees reported positive responses to the genetic result from the start. Others initially had negative experiences upon receiving the result, but subsequently experienced more positive feelings about receiving the result once they had a chance to process the result and seek out additional information. Positive responses included feeling “grateful” and concluding that the result was useful for informing future healthcare (such as getting mammograms earlier) and for informing or altering their own health behaviors to “live a healthier lifestyle” or “eat right and exercise.” Other positive responses included feeling cared for by the healthcare system and knowing that “they [the healthcare system] are going to be keeping a close eye on everything.”

Interviewees also reported that increased awareness could help them prepare and educate themselves, and that they felt proactive regarding themselves and their families:


*“… a couple of weeks ago they did find a tumor in my stomach. So, I’m trying to talk my other relatives into being tested and my child, and my brother…I tried to explain to them…It just makes you more aware of what could possibly be going on. I found it interesting and thought it was something that I should do.” (Cascade)*



*“…I knew that I had high cholesterol, I had heart disease. It’s rampant in our family. But, what it did for me was we could then pass it on to other people in the family, that it was genetic. And I have a nephew who is young, I think about 10, and they checked his cholesterol and it’s high already…it’s a good thing that that was investigated.” (FH)*


Many reported that while they might have been surprised by the result, they were intrigued by the information and it felt like a “relief,” due either to knowing the cause of the specific health problems that they or their family members have already dealt with, or to having a more definitive answer for their health concerns (objectively related to the condition or assumed to be related in the interviewee’s perception). Some interviewees reported receiving support from their spiritual and religious beliefs as well as from others, including family and friends, who facilitated efforts to cope with the result. A few interviewees noted feelings of gratitude to the MyCode program for contributing to society and research.

Some interviewees noted more neutral reactions such as not being surprised by the result due to information that they already had, such as a family history. Others reported that not much changed due to the result, either because they felt like there was nothing they could do, or they felt they were already doing everything possible with their current healthcare and through personal actions. Neutral reactions were also seen in older interviewees:


*“… we’ve got a lot of other issues at our age. There are probably other things that are going to knock us off this world more than what anything this will affect us, you know.” (FHxNeg)*


Some reported that the result was not a “big deal,” either because they did not have any specific symptoms yet or because they had other health problems that were more concerning or more challenging to manage at the moment:


*“Well, it’s the last thing on my mind. I have other more serious health issues. I was diagnosed recently with ALS, and so the last thing on my mind is the BRCA1 gene.” (FHxNeg)*


Some interviewees reasoned that having health problems and facing mortality are inevitable. Others reported that they were not worried; some voiced confidence in healthcare professionals. Some stated that they did not feel differently (in terms of their health); others spoke of being “pragmatic.” Some noted that the genetic result was no different from other health-related information they receive:


*“Well, the other medical information has been pretty generic. You’re old. You’re fat. You need to lose weight, things like that. The genetic testing is a little bit more personal, but it wasn’t upsetting. You know, it’s an indicator. It’s not a guarantee. So, it didn’t raise concern. It just raised awareness.” (PEX)*


Interviewees without a family history of the condition often indicated a need for a period of adjustment not expressed by interviewees who had a family history. One interviewee reported that she was “shocked” to learn that she had a higher risk for breast cancer due to a *BRCA* result, given that she had no family history of this. She received the information at age 33 during a pregnancy and, as a result, had a stressful experience. Despite being “shocked, upset, and stressed” about receiving the result and sharing that information with her family, she also reported that her feelings changed over time:


*“I’m glad I know. I would rather know this way than by finding out I had cancer…so this is ‘the best-case scenario’.” (FHxneg)*


Another interviewee expressed a negative reaction to receiving a *BRCA* result initially:


*“I didn’t really feel that there was a strong family history of breast cancer, so when it came back positive, I was kind of like, wow, how unfortunate…you know, why am I the one?”*


She then described several steps she went through to manage these negative emotions, including meeting with a genetic counselor, beginning breast cancer surveillance, having risk-reducing salpingo-oophorectomy, and confiding in her husband and aunt:


*“I met with the genomics team…it went well. I turned to my husband, and I turned to my aunt, and they were very supportive. They’re my go-to people. I think I’ve been pretty positive, knowing now I got it and I’m going to deal with it, and I got it covered, so it will be okay.” (PEX)*


### 3.3. Behavior Change Domain

This domain captured whether interviewees discussed making behavior changes of any kind as a result of the genetic information received. Also included here is a discussion of barriers to obtaining services or making changes. Interviewees reported a variety of actions taken, which also varied depending on individual context. The most common action taken was consultation with a clinician (e.g., primary care physician, genetic counselor, cardiologist (FH, HCM), or inherited risk breast clinic specialist (HBOC)). The next most commonly reported action taken was disease risk management. Some interviewees reported having had prophylactic surgery, while others had scheduled an appointment for risk management but had not yet performed any risk management procedures.

Among interviewees who reported condition-specific, recommended risk management, half clearly stated that the risk management was due to the result. For the remainder, it was unclear whether the risk management was specifically related to receiving their genetic result or to the fact that they were already engaged in actions relevant to the result as part of regular average-risk screening recommendations (e.g., cholesterol medication or mammograms):


*“Now I’ve accepted it, and we do all the preventatives. I, you know, take my pills. I go in every 6 months for a mammogram, 6 months later for an MRI, 6 months later for oncology, and I’ve accepted it” (FHxNeg)*



*“I was always pretty good… before this. Like going for the mammograms and having the checkups, so really for me it didn’t change anything.” (FHxNeg)*


Sharing the result with family was another action commonly reported, as was information seeking, often from the internet. A few interviewees reported medication changes and lifestyle changes, although it was not always clear that the lifestyle changes (which usually concerned diet) were related to the genetic result.

Reported barriers to acting on their result included cost and insurance:


*“If he [interviewee’s husband] were not working, … I would not pursue it probably as actively. I would hold off. I definitely would hold off.” (PEX)*


A few interviewees reported no actions or no information seeking beyond the initial disclosure or genetic counseling appointment. One interviewee provided insight, saying:


*“I don’t really think about it, how it’s gonna affect me. I think there are other things that are gonna take me out of the picture long before any of… I could be wrong, 100% wrong, you know, but like I said, I’ve got more things that are on my mind than something like that” (FHxNeg)*


### 3.4. Family Communication Domain

Interviewees indicated that they shared results with family members and provided details about what they know about the health implications of their result, their own assessments of family members’ risk, family dynamics affecting communication, and family response to learning about the genetic information. This included how relatives responded emotionally and what actions those family members did, did not, or planned to take.

Most interviewees reported sharing the information with at least one family member. However, the nature of the information, with whom they shared it, and why, varied. Some interviewees reported choosing not to give information to some family members. Choice of with whom to share information was based on such factors as geographic distance or emotional closeness. Others chose to share with certain relatives but not others based on their understanding, or misunderstanding, of the genetic information. For instance, several interviewees who received a *BRCA1/2* result indicated that they only needed to inform female relatives. Likewise, among individuals with FH results, multiple interviewees appeared to report discussing “high cholesterol” within the family but not the genetic basis and heritability of FH specifically.

Interviewees who shared their result also expressed not having influence over family members’ actions. One interviewee reported a belief that the information did not spur his family members to action because cancer is no longer “devastating news to anybody.” Interviewees who were family members (i.e., were in the Cascade cohort) and had submitted a sample for cascade testing reported doing so because they believed that the health information could be valuable for them or their families.

## 4. Discussion

We aimed to understand the lived experiences—the responses—of MyCode participants and their family members who have received clinically actionable genomic information in a population screening context. The study was conducted among a large, qualitative sample purposively selected to include the experiences of individuals with and without objective evidence of personal or family history related to the result, individuals who received a result associated with risk for cancer or cardiovascular disease, and family members who completed cascade testing for a familial variant. Our results, which are consistent with findings from research on reactions to clinical testing and early research on secondary findings, provide important context for population screening where actionable genetic information is provided as a screening or prevention tool regardless of clinical indication. Overall, participants reported a positive or neutral psychological response, with most negative reactions being manageable using available resources; they felt that receiving the results was an important part of their healthcare and that they would use the information to guide their behavior.

Providing actionable genetic information as a screening tool regardless of clinical indication and the process used to disclose results appeared to be acceptable for all interviewees. Although there was disagreement about the clarity of the letter summarizing the result, this is consistent with known challenges involved in presenting written information to participants with different levels of educational attainment and scientific/health literacy [30]. While these findings are specific to MyCode, similar letters are used by most clinical programs and by other research studies returning genetic results to individuals from a screening context [30].

Psychological responses varied and appeared to depend on individual context, population sample group, and condition, especially for initial responses to receiving the result. In our sample, interviewees without a personal or family history they perceived as consistent with the genetic result used some of the strongest language to express their shock, disbelief, and disappointment at receiving the result. Yet, these interviewees appeared to adjust to their initial negative feelings by using sources of support available to them (e.g., managing risk via medical procedures or confiding in family). Our findings support the growing evidence that patients view and respond to genetic information similarly to other health information, suggesting genetic results need not be treated as exceptional [31]. The observation that attitudes changed over time also fits well with the transactional model of stress and coping [32], in which participants’ initial experiential appraisal is based on personal factors such as previous knowledge or risk. Those who experienced an initial negative reaction appeared to go through a secondary appraisal where factors such as resources, problem-focused coping, and social support helped mitigate the stress response, leading to adjustment over time [32]. This is consistent with research carried out in a diagnostic genetic testing context showing those receiving a positive result typically have an increase in distress that returns to baseline usually over a 6–12-month period of time [33]. Additionally, it suggests that genomic screening programs can provide a supportive infrastructure for disclosure and risk management (e.g., via psychologists, genetic counselors, or specialists who perform risk management procedures) to help mitigate any initial negative reactions. Further, it suggests that ongoing support may be required as some patients adjust to the genetic result.

It is important to note that some responses indicated the genetic information is of lower priority compared to other health issues that the participants were already experiencing. This response may represent a key difference in the population screening context compared with clinical testing. In clinical testing, patients are specifically engaging in genetic testing due to suspicion of a genetic condition and may, therefore, find the result more impactful, whereas in the population screening context this information is immediately integrated into the context of other life and health issues.

This study adds to recent evidence [3,8] that most patients do not experience unmanageable, ongoing psychological distress. However, in spite of this mostly favorable reaction, other research from the MyCode program demonstrates that there is significant room for improvement in adherence to post-disclosure risk management among those who received a genetic result through population screening [2].

Post-disclosure behaviors reported by interviewees in this study further support prior case reports in this population indicating that disclosure of genetic variants may spur risk management [34,35] or follow-up on previously ignored screening results [36]. This information could inform future research on increasing adherence to guideline-based care [22,37,38]. Although not formally assessed, applying stages of change models to interview transcripts, participants within this analysis likely fall into the contemplation or preparation stage (i.e., information seeking and planning or scheduling appointments) or the maintenance stage (i.e., engaging in regular surveillance) [39], suggesting that interventions such as motivational interviewing could help those who are ambivalent or unsure of how best to follow up or engage in health management related to their genetic result [37]. Barriers reported by our respondents were cost- or insurance-related; simple problem solving delivered by clinicians could help individuals navigate these types of issues and increase adherence to recommended treatment and risk management.

Results in the family communication domain are consistent with prior reports that patients have varied experiences of sharing information with family members and that the outcome of that communication is likewise varied [8,40,41,42,43]. As in clinical testing [44], emotional and geographic closeness to family members and understanding the result and its implications seemed to play a role in whom participants chose to notify. Importantly, this indicates that to patients, genetic ties and familial ties are not synonymous. Cascade testing counseling must account for this distinction. Although sharing information is crucial to improving cascade testing rates, family members must then decide whether to engage in cascade testing and likely experience the same barriers as those identified in previous research [45,46]. Thus, information sharing alone may not be sufficient to improve cascade testing rates and is likely a poor proximal measure for uptake of cascade testing. There is evidence that cascade testing uptake among relatives of individuals who received a genetic result through MyCode, or through other similar studies, has been poor [47] despite mostly positive or neutral reactions [3,8]. Therefore, new interventions informed by behavioral science must be developed and tested if the promise of improved health and disease prevention through provision of genetic information is to be realized at the individual, family, and population levels [48,49].

## 5. Limitations

It is possible that study participants’ experiences do not cover the gamut of experiences of individuals who receive results via a genomic screening program. Study participants made up a subset of individuals who received a result via MyCode, which comprised individuals who, by virtue of consenting to MyCode, might not be representative of the overall Geisinger patient population. Future studies could assess the impact on patients’ experiences of additional factors not collected for this study (e.g., genetic counseling content, education level, sociodemographic characteristics, or health insurance coverage). Additionally, because our sampling strategy included multiple groups of individuals with different genetic results and varied times since receiving results, only limited comparisons among interviewees are possible despite what would be considered a large sample size for qualitative research. Rather, this purposive sampling of individuals was specifically designed to determine the breadth of patient experience across several different contexts to better inform the rapidly increasing number of programs returning genomic information from population screening. The inclusion of individuals from different condition groups, individuals with and without family history, and family members choosing cascade testing further provides guidance across contexts and highlights shared and divergent experiences across these contexts. Inherent in all research of this type is the inability to assess the experience of those who decline to participate, which could bias the final interpretation.

## 6. Conclusions

We aimed to understand the depth and breadth of patient experiences after receiving genomic information from a population screening initiative. Our findings augment the mounting evidence that patients view and respond to genetic information similarly to other health-related information and feel genetic results are useful to inform care. These findings provide guidance for the growing number of programs that use, or plan to use, genomic screening. Findings also illuminate opportunities to improve clinical utility through further research and interventions targeting adherence to risk management recommendations following a result, family communication, and cascade testing.

## Figures and Tables

**Table 1 jpm-12-01511-t001:** Qualitative interview questions by study cohort.

		Study Cohort			
Question Purpose	General Question *	PEX	FHxNeg	FH	Cascade
Opening/patient story	Tell me about your experience learning about the MyCode genomic information.	X	X	X	X
Result follow-up/medical care	What have you done since finding out about this result?	X	X	X	
Communication about result	How has your family responded to this result?	X	X	X	X
Understanding of information and resource seeking	How do you describe this result to others?	X			X
Results disclosure processes	What are your thoughts on the processes for finding out about the result?	X	X		
Psychological reactions to result	Please describe any effect these results had on your feelings.	X	X		X
Financial implications	Do you have any concerns about paying for medical care needed because of the result?	X			
Satisfaction with decision to participate in MyCode	Now that you have received this information, how do you feel about your decision to participate in MyCode?	X			X
Decision to test for familial variant	What most motivated you to have genetic counseling and testing?				X
Participant characteristics	sex, age, insurance, race/ethnicity, health literacy, employment, income	X	X	X	

* question language adapted to be appropriate to interviewee population group; PEX = post-result experience; FHxNeg = family history negative for disease associated with genetic result; FH = familial hypercholesterolemia; Cascade = considered cascade testing for familial genetic variant.

**Table 2 jpm-12-01511-t002:** Cohort characteristics.

Cohort	Interview Dates	Identified as Eligible	Unable to Contact	Declined Participation	Completed Interviews
**PEX**	May–August 2016	67	20	18	29 *
					(HBOC = 18; LS = 5; Long QT = 2; HCM = 2;
					FH = 3)
**FHxNeg**	March–April 2018	45	13	14	19
					(HBOC = 13; HCM = 6)
**FH**	July 2017	26	13	6	7
**Cascade**	May 2017	26	13	5	8 **
					(HBOC = 5; LS = 2; FH = 2; PGL-PCC = 1)
**Total**		164	59	43	63 ***

HBOC = hereditary breast and ovarian cancer syndrome; LS = Lynch syndrome; HCM = hypertrophic cardiomyopathy; FH = familial hypercholesterolemia; PGL-PCC = hereditary paraganglioma-pheochromocytoma syndrome; * 1 participant had a pathogenic variant in both the *BRCA2* and *APOB* genes; ** 1 participant was tested for familial pathogenic variants in the *BRCA2* and *MSH2* genes; 1 participant was tested for familial pathogenic variants in the *BRCA2* and *APOB* genes; *** 60 participants completed 63 interviews; 2 completed interviews in the PEX and FHxNeg cohorts; 1 completed interviews in the FHxNeg and FH cohorts.

## Data Availability

The data presented in this study are available on request from the corresponding author. The data are not publicly available due to the potential to identify participants from qualitative data.

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
