# Peer review of "Understanding the Patient Experience of Receiving Clinically Actionable Genetic Results from the MyCode Community Health Initiative, a Population-Based Genomic Screening Initiative"

_jpm, 2022, doi:10.3390/jpm12091511_

Round 1

Reviewer 1 Report

The manuscript "Understanding the patient experience..." highlights important questions relevant as genomic screening in different settings is implemented and secondary findings revealed. It is therefore of great importance that we as clinicians understand and are aware of the implication of this screening for our patients/relatives and that we recognize the gaps we might have in the design of taking care of them adequately after receiving the results. Therefore this study is of great value and gives some preliminary answers and what we need to follow up.

Here some suggestions and small adjustments and also som questions; some might be because I have not fully understood and if, my apologies.

Introduction: The sentence: Data were collected from -structured....Could this sentence be revised so it is more clear, is it the population screening that is studied or the result from them or something else? 

Material and methods section: Interview process: Here I am not clear about what you mean in the sentence: Interview questions were informed reactions to genetic information.... Probable I have not enough background knowledge about methodology or maybe language, on the other hand I think your messages is important and it would be important to understand also for someone not familiar with this type of research. Could this be clarified?

Table 1: Although the abbreviations are explained in the text, it would be helpful to either refer to the textsection or to have the abbreviations as an underlying text.

Questions: How was question language adapted in the interviews ?

Did the cascade cohort get more information than what was given the familyindex, i.e., did they have more thorough genetic counselling affecting how they reacted?

The Sampling strategy: You describe that you sampled groups with different characteristics compared to those your already interviewed (and the already interviewed are also part of this study?).

Results: Just curious, from how many of totally screened did you sample your cohort of secondary findings?

Finally just some reflections to consider and that could be included in the discussion:

You have a comment about financial and insurance considerations-these might be different in another health care system, i.e. reactions might differ when you have to be afraid that you might not be able to afford the care suggested?

You also mentioned the bias in selection of those who participate in this kind of a population study. Do you have any possibility to compare, for instance education level, religious background, age/sex distribution, and socioeconomic factors to the general population. These factors could affect how someone reacts to the results?

Communication in the family-did it differ in different groups (education level, socioeconomic, culture, age, sex). You discuss this with respect to BRCA 1/2 and also FH, but how about other factors than the disease itself. You also have a sentence about that some chose not to inform, does these interviewees differ from others.

You discuss that the results have changed the behavior; but how is empowerment, or maybe this is difficult to evaluate from these questions? How do we know they change it for real in the long run?

You also discuss that the genetic information was for some considered less important; were they older? Did they know less about genetics or was it something else?

Overall, a very interesting paper, raising many more questions. I do not expect that all my questions can be answered, but it would be interesting to hear your comments. Also, what are your next steps, any ideas for improvement how we should do differently as clinicians and clinics?

Author Response

We would like to thank Reviewer 1 for their comments, which highlight important questions relevant to recently implemented genomic screening settings. We appreciate the suggestions and have made the following revisions:

  1. The manuscript "Understanding the patient experience..." highlights important questions relevant as genomic screening in different settings is implemented and secondary findings revealed. It is therefore of great importance that we as clinicians understand and are aware of the implication of this screening for our patients/relatives and that we recognize the gaps we might have in the design of taking care of them adequately after receiving the results. Therefore this study is of great value and gives some preliminary answers and what we need to follow up.

We thank the reviewer for the kind words regarding the value of this study.

  1. Introduction:The sentence: Data were collected from -structured....Could this sentence be revised so it is more clear, is it the population screening that is studied or the result from them or something else? 

We have added a clause to the sentence in question (p. 4) to note that what we studied were the post-disclosure experiences associated with receiving a genetic result through the genomic screening program.

  1. Material and methods section: Interview process: Here I am not clear about what you mean in the sentence: Interview questions were informed reactions to genetic information.... Probable I have not enough background knowledge about methodology or maybe language, on the other hand I think your messages is important and it would be important to understand also for someone not familiar with this type of research. Could this be clarified?

We have replaced the jargon of “deontological conjectures” with a phrase noting expert opinion as the source of theoretical reactions to genetic results received via genomic screening (pp. 5-6).

  1. Table 1:Although the abbreviations are explained in the text, it would be helpful to either refer to the text section or to have the abbreviations as an underlying text.

We have added definitions of the abbreviations below Table 1.

  1. Questions: How was question language adapted in the interviews?

We have added an example to the two existing parenthetical examples of ways in which interview questions were adapted for the study cohorts (p. 6).

  1. Did the cascade cohort get more information than what was given the family/index, i.e., did they have more thorough genetic counselling affecting how they reacted?

Although it is plausible that the thoroughness of genetic counseling differed by participant on factors such as whether they had cascade genetic counseling vs. proband genetic counseling, we did not audit genetic counseling sessions or collect other data that could have answered this question. We have added a sentence to Discussion, Limitations to suggest future research that considers the impact of such factors on patients’ experiences (p. 21).

  1. The Sampling strategy:You describe that you sampled groups with different characteristics compared to those your already interviewed (and the already interviewed are also part of this study?)

As described in Methods, this study was conducted in stages over two years. We conducted each successive stage of interviews to determine whether new themes emerged in groups that differed from previous groups in ways that could influence their reactions to results (e.g., different genetic diseases, different family history of relevant diseases). Those interviewed at each stage are included in this study.

  1. Results:Just curious, from how many of totally screened did you sample your cohort of secondary findings?

As noted in Table 2, the final study cohorts were drawn from an eligible sample population of 164 individuals.

Finally just some reflections to consider and that could be included in the discussion:

  1. You have a comment about financial and insurance considerations-these might be different in another health care system, i.e. reactions might differ when you have to be afraid that you might not be able to afford the care suggested?

We agree and believe it is critical to study coverage of risk management across diverse health systems and types of insurance. The sentence added to the Discussion, Limitations (see response to #6) is intended to address this point.

  1. You also mentioned the bias in selection of those who participate in this kind of a population study. Do you have any possibility to compare, for instance education level, religious background, age/sex distribution, and socioeconomic factors to the general population. These factors could affect how someone reacts to the results?

We agree with the reviewer that reactions to results could be influenced by a variety of participant characteristics. Unfortunately, we did not collect the suggested participant characteristics for this study. We have noted this limitation in Discussion, Limitations and suggested that future research include consideration of such factors.

  1. Communication in the family-did it differ in different groups (education level, socioeconomic, culture, age, sex). You discuss this with respect to BRCA 1/2 and also FH, but how about other factors than the disease itself. You also have a sentence about that some chose not to inform, does these interviewees differ from others.

As noted above (see response to #10), we did not collect some of these participant characteristics for this study and have highlighted doing so as an opportunity for future research.

  1. You discuss that the results have changed the behavior; but how is empowerment, or maybe this is difficult to evaluate from these questions? How do we know they change it for real in the long run?

This is an important question that is beyond the scope of this manuscript. We have ongoing research on adherence to recommended risk management for these genetic conditions.

  1. You also discuss that the genetic information was for some considered less important; were they older? Did they know less about genetics or was it something else?

We cannot definitively say why the genetic information was considered less important for each participant who reported lower perceived importance. However, as highlighted in the exemplar quotes on p.13, some participants reported that other health risks presented more pressing concerns than the genetic result.

  1. Overall, a very interesting paper, raising many more questions. I do not expect that all my questions can be answered, but it would be interesting to hear your comments. Also, what are your next steps, any ideas for improvement how we should do differently as clinicians and clinics?

Thank you for the kind words and thoughtful questions. We are currently using a quantitative approach to study reactions to genetic results (e.g., positive and negative emotions, FACToR subscales, decision regret) and are developing electronic health record-based approaches to assessing adherence to recommended risk management over time. Designing clinical programs that report genetic results such that they collect implementation outcomes in standardized fashion will be critical to understanding the clinical utility of genomic screening in unselected populations.

Reviewer 2 Report

Overall I found this paper to be interesting and quite relevant to the field. While the experience of individuals in the Geisinger system may have a different experience from those in another healthcare system, the data presented here are consistent with previous studies supporting that it is more broadly applicable. While there are many other areas that could be explored in this strain, like adaptation to return of results with a GC v. a physician or the impact of reclassification of variants, this addresses the primary issues presented. I didn't have any major comments or changes. , 

Author Response

We would like to thank Reviewer 2 for their comments and agree that these results are relevant and broadly applicable.

Reviewer 3 Report

The Article "Understanding the patient experience of receiving clinically actionable genetic results from the MyCode Community Health Initiative, a population-based genomic screening initiative" reports patients experiences after receiving genomic information. There is no proper scientific statistic analysis and the cohorts are small which make the conclusion inconvincible.

Author Response

Qualitative methodology like that used in this manuscript is a useful and well-recognized tool in research, particularly for hypothesis generation during early stages of research or when understanding of experiences or observations is required. The references below, which have been cited collectively more than 170,000 times, are just a few of the many references to support the approach we took. For this project, qualitative analyses provided us with a greater understanding of patients’ experiences after receiving a clinically actionable result from a research biobank, a method of identifying and disclosing genetic risk that is increasingly being implemented at healthcare systems across the United States. In addition, the number of participants in the cohort was typical of, if not greater than, a qualitative study and was informed by the principles of purposive sampling for like experiences and saturation. When using this methodology, participants with specific experiences are enrolled until no new information is reported across different participant characteristics.

Timmermans, S. and I. Tavory, Theory Construction in Qualitative Research: From Grounded Theory to Abductive Analysis. Sociological Theory, 2012. 30(3): p. 167-186.

Patton, M.Q., Qualitative evaluation and research methods. 2nd ed. 1990, Newbury Park, CA: Sage.

Landis, J.R. and G.G. Koch, The measurement of observer agreement for categorical data. Biometrics, 1977. 33(1): p. 159-74.

Reviewer 4 Report

This manuscript deals with an interesting topic but has some significant limitations. The breadth of the approach, including several diagnoses with different severity limits the conclusions that can be drawn. It also reduces the number of individuals in each category (and some statistics would be appropriate to show/refute the point). The presentation of the data is too much "talk" and less/no graphical presentation which markedly reduces the value of their manuscript. How many individuals were "positive, neutral or negative and which disease genes were they carrying. 

Author Response

Please refer to the response above addressing that qualitative methodology was an appropriate methodology to use for this research and that this research was conducted rigorously based on the standards for this approach. Exemplar quotes and thematic analyses are the accepted and most useful way of presenting research findings from qualitative interviews. Presenting data as counts of positive, negative, and neutral valences would oversimplify the nuances of experiences, including changes in these experiences over time. The approach of using purposive sampling to achieve study cohorts that differed from one another in factors that could influence reactions to genetic results is a strength of the study’s methodology, not a limitation. It allowed us to better understand the breadth of individual experiences with this new and expanding type of genetic information delivery. And, we found that many of the reported themes and reactions were common across study cohorts, an important finding of the study.

Round 2

Reviewer 3 Report

My comments have been addressed well.

Author Response

My comments have been addressed well.

We are pleased that the response addressed the reviewer’s comments.

Reviewer 4 Report

I would still like to see the details I asked for last time but if the authors claim that this is standard procedure in their field, I surrender, but I would then like them to include the statement that this is indeed standard procedure.

In my view, this is still "soft" science where its scientific value would be enhanced by including some degree of numerical evaluation.

Author Response

I would still like to see the details I asked for last time but if the authors claim that this is standard procedure in their field, I surrender, but I would then like them to include the statement that this is indeed standard procedure.

In my view, this is still "soft" science where its scientific value would be enhanced by including some degree of numerical evaluation.

We have added to Methods, Data analysis (p. 8) a sentence stating that we presented study data as themes and illustrative quotes, according to standards for rigorous qualitative inquiry. The sentence is supported with references to these methodology standards.